# Statins in Children, an Update

**DOI:** 10.3390/ijms24021366

**Published:** 2023-01-10

**Authors:** Riccardo Fiorentino, Francesco Chiarelli

**Affiliations:** Department of Paediatrics, University of Chieti, Via dei Vestini, 66100 Chieti, Italy

**Keywords:** dyslipidemia, statin, children, adolescents

## Abstract

Since lipid abnormalities tend to progress from childhood to adulthood, it is necessary to early identify and treat children and adolescents with dyslipidemia. This is important in order to reduce the cardiovascular risk, delay the development of fatty streaks, slow the progression of atherosclerosis and reverse atherosclerotic plaques. Together with therapeutic lifestyle changes, statins are the most common lipid-lowering drugs. By inhibiting the endogenous cholesterol synthesis in the liver, statins increase the catabolism of LDL-C, reduce VLDL-C, IDL-C and TG and modestly increase HDL-C. Regardless of their lipid-lowering effect, statins have also pleiotropic effects. Statins have increasingly been prescribed in children and adolescents and mounting evidence suggests their beneficial role. As with adults, in children, several studies have demonstrated that statin therapy is efficient at lowering lipid levels and reducing CIMT progression and cumulative estimated atherosclerotic burden in children. Statins are generally very well-tolerated in both adults and children and adverse events are quite uncommon. When evaluating the need and the timing for statin treatment, the presence of several factors (secondary causes, familial history, additional risk factors) should also be considered. Before initiating statins, it is imperative for clinical practitioners to consult patients and families and, as with any new medication therapy, to monitor patients taking statins. Despite being safe and effective, many children with lipid disorders are not on statin therapy and are not receiving the full potential benefit of adequate lipid-lowering therapies. It is therefore important that clinicians become familiar with statins.

## 1. Introduction

Atherosclerosis is a chronic inflammatory condition, characterized by the build-up of plaques inside the arteries and the progressive thickening of the arterial wall [1]. Despite a long initial asymptomatic phase, it is known that atherosclerosis is a potentially serious disease. The progression of the insidious atherosclerotic process may lead to cerebrovascular disease, coronary events and other atherosclerotic cardiovascular diseases (ASCVD), which are currently the leading cause of morbidity, disability and premature death worldwide [2,3]. The atherosclerotic process is mainly driven by lipids and initiates with the deposition of lipids on the inner lining of arteries. Lipid accumulation over time leads to an endothelial dysfunction and activates pro-inflammatory processes, which induce the formation of fatty streaks (the earliest atherosclerotic lesions) and their gradual evolution into atherosclerotic plaques [4,5]. Although clinical outcomes of atherosclerosis are usually observed in adults, several studies have confirmed that the pathological process begins much earlier [6]. Landmark studies showed that fatty streaks were found in the coronary arteries of children as young as 2 years of age [7]. More recently, it has been described that children with traditional cardiovascular risk factors had increased carotid intima–media thickness (CIMT), which is a surrogate biomarker of atherosclerosis [8]. In addition, recent studies have show that coronary intimal thickening may begin in utero [9,10]. Therefore, these results suggest the precocious origins of adult diseases and allows us to consider atherosclerosis as a significant problem in children. Several cardiovascular risk factors and conditions contribute to atherosclerosis in children: among them, childhood dyslipidemia plays a key role [11]. Indeed, it is well known that lipids are essential for cellular organization, stability and functionality as well as for steroid hormones, vitamins and bile acids synthesis [12]. However, lipid and lipoprotein abnormalities can be detrimental to the human body and are linked to the initiation and progression of atherosclerosis in children [13]. In particular, higher levels and longer exposures to increased low-density lipoprotein (LDL-C) are the main factors that exacerbate the atherosclerotic progression [14]. Since lipid abnormalities tend to progress from childhood to adulthood, it is necessary to early identify and treat children and adolescents with dyslipidemia. This is important in order to reduce the cardiovascular risk, delay the development of fatty streaks, slow the progression of atherosclerosis and reverse atherosclerotic plaques, whenever possible. Together with therapeutic lifestyle changes, statins are the most common lipid-lowering drugs and represent the cornerstone of primary and secondary prevention of cardiovascular diseases in adults [15,16]. Statins were increasingly prescribed in children and adolescents and mounting evidence suggested their beneficial role in children and adolescents. This narrative review aims to highlight the efficacy, safety and rationale for the use of statins in pediatrics. Moreover, we summarize the indications for clinical use of statin therapy, focusing on initiation and monitoring of treatment in children and adolescents.

## 2. Cholesterol Metabolism and Mechanism of Action for Statins

All cholesterol in the human body comes from two major sources: it can be either obtained through diet (20% of total cholesterol) or synthetized de novo (80% of total cholesterol). Although all cells are capable of synthesizing cholesterol, the major cholesterol production occurs in the liver [17]. Cholesterol synthesis is a complex process; several enzymes are involved, and the biosynthetic pathway is tightly regulated at several points [18]. Cholesterol production starts with the formation of mevalonate from acetate; subsequently, mevalonate is transformed to squalene (the biochemical precursor of all steroids) in the endoplasmic reticulum. Squalene is further converted to lanosterol and finally transformed into cholesterol [12,18]. Newly synthesized cholesterol then must leave the endoplasmic reticulum to exert its functions: through different mechanisms, it moves to the membranes and is transported to peripheral tissues. However, due to its insolubility in plasma, cholesterol is required to be packaged with lipoproteins: lipoproteins are spherical macromolecules, consisting of a hydrophobic core (containing cholesterol esters and triglycerides) and a hydrophilic coat (containing free cholesterol, phospholipids and apolipoproteins) [19]. There are several lipoproteins and different lipoprotein pathways. Via the exogenous pathway, dietary lipids are incorporated in chylomicrons, transported from the intestine to peripheral tissues and then taken up by the liver; via the endogenous pathway, lipids are transported from the liver to peripheral tissues [20]. The endogenous pathway begins in the liver with the production of very-low-density lipoproteins (VLDL-C), which contain cholesterol and a large amount of triglycerides (TG). In the circulation, the triglycerides carried in VLDL-C are hydrolyzed by lipoprotein lipase (LPL), resulting in the formation of intermediate-density lipoproteins (IDL-C) and the further conversion of IDL-C to low-density lipoprotein (LDL-C). LDL-C (the major cholesterol-rich lipoproteins) then reach peripheral cells and are cleared by circulation through the interaction with LDL-C receptor, which is present in many tissues and most abundant in the liver [19,20]. Lastly, through the reverse transport pathway (reverse cholesterol transport) the organism removes the excess fats from peripheral cells and transports them to the liver: the process is mediated by high-density lipoproteins (HDL-C), which exhibit various anti-atherogenic and cardioprotective effects [21]. Statins, also known as hydroxymethylglutaryl-CoA (HMG-CoA) reductase inhibitors, are a class of lipid-lowering drugs that work by inhibiting the endogenous cholesterol synthesis in the liver. In particular, statins are potent competitive inhibitors of HMG-CoA reductase [22]. HMG-CoA reductase is involved in the final step of the mevalonate pathway and catalyzes the reaction where HMG-CoA is reduced to mevalonate (NADP-dependent synthesis of mevalonate). This enzymatic reaction is not only an irreversible and tightly regulated step, but also the rate-limiting one of cholesterol biosynthesis [23]. On the basis of their structure, there are two different types of statins: type 1 statins (pravastatin, simvastatin and lovastatin) are products of natural origin, whereas type 2 statins (atorvastatin, fluvastatin, pitavastatin, rosuvastatin) are synthetic products that are characterized by attached fluorophenyl groups and larger hydrophobic regions [18]. The capacity of statins to inhibit HMG-CoA reductase is due to the presence of an HMG-like moiety that binds to the HMG binding site. Moreover, the hydrophobic region of type 2 statins contributes to block the binding of HMG-CoA to the active site [18,24]. By competitively blocking the HMG-CoA reductase, statins alter the endogenous pathway and reduce the intrahepatic cholesterol amount: consequently, this promotes the hepatic upregulation of the LDL-C receptor (increased synthesis and expression) and increases the catabolism of circulating LDL-C. In addition, statins reduce VLDL-C, IDL-C and TG and modestly increase HDL-C (improving the reverse transport pathway) [25]. Lastly, regardless of their lipid-lowering effect, statins have pleiotropic effects on endothelial function, inflammation, coagulation and oxidative stress (Figure 1) [26,27].

## 3. Efficacy

It is well known that statins are effective in adults: in addition to their variable lipid-lowering effect, statins are beneficial in improving the lipid profile, conferring cardiovascular protection (in terms of primary and secondary prevention) and reducing cardiovascular morbidity and mortality [22,23]. Over time, statins have increasingly been used in children and adolescents and several studies and meta-analyses have evaluated the efficacy of statins with children and adolescents [28,29,30]. Although most studies have been conducted on patients with pediatric familial hypercholesterolemia (FH), they highlighted the beneficial effect of different statins in youths with dyslipidemia [31,32]. As with adults, early studies demonstrated that statin therapy is efficient at lowering lipid levels in children: despite a variable efficacy through the trials, all statins led to a significant reduction in LDL-C levels when compared to placebo [14,33,34]. The differences in the mean lipid-lowering effect were probably due to different baseline LDL-C values, statin dosages and formulations [29]. In fact, it is important to remember that the lipid-lowering effect of statins changes when the dose of HMG-CoA reductase inhibitors increases (dose-dependent effect) and when a high-potency class of statins is used. Because of the longest terminal half-life, rosuvastatin, pitavastatin and atorvastatin are the most potent statins; on the other hand, lovastatin, pravastatin and fluvastatin are the least potent; simvastatin is a moderately potent statin [34]. In accordance with recent studies, HMG-CoA reductase inhibitors in children led to a relative reduction in LDL-C values by approximately 21–41% (mean relative decrease of 32%) [28,35]; moreover, thanks to statin treatment, the majority of children might achieve the desired LDL-C goals [30,36]. In addition, a recent meta-analysis demonstrated that statins were helpful in reducing total cholesterol and TG (by 25% and 8% respectively) and increasing HDL-C (by 3%), when compared with placebo [30]. As mentioned above, the use of statins in adults significantly reduces cardiovascular morbidity and this also applies to patients with FH; in adults with FH, statins may significantly reduce the ASCVD rates and delay the median age of onset of cardiovascular events in comparison with the pre-statin era [14,37]. In children, the results seem very similar: by performing a 20-year follow-up study, a recent study aimed to compare the incidence of cardiovascular events and death in patients with FH who initiated statins in childhood and in their affected parents who initiated statins much later in life. Very interestingly, after 20 years, the initiation of statin therapy at an age of 8–18 years resulted in the significant reduction of ASCVD rates and death from cardiovascular events. Moreover, the cumulative cardiovascular disease-free survival was significantly higher among patients who received statins [35]. Although these findings are particularly surprising, it is important to note that the study is only observational and other studies are needed to confirm the results. Nowadays, however, randomized control trials designed to measure the effects of statins during childhood on cardiovascular outcomes are lacking; since cardiovascular events rarely occur in children, there are probably many challenges associated with the conception of event-driven trials. Therefore, several studies have tried to evaluate the effects of HMG-CoA reductase inhibitors on surrogate markers of atherosclerosis. In particular, the CIMT represented the most evaluated noninvasive measure of early atherosclerosis [38]. In a landmark trial (2004), children with FH were randomized to pravastatin or placebo: over a two-year treatment period, the CIMT was significantly greater in those receiving a placebo. Moreover, trends toward CIMT progression and CIMT regression were found in the placebo and the statin-treated group, respectively [39]. The placebo-controlled phase was then followed by an open-label extension: some authors demonstrated that the earlier initiation of pravastatin was associated with a lower CIMT following approximately 4 years of treatment [40]. Most recently, a 10-year follow-up study found that, before starting the treatment, children with FH showed a greater CIMT when compared with non-FH controls; however, when statin therapy was initiated, important improvements in CIMT (reduced CIMT, slowed progression) were observed: very interestingly, the CIMT progression rates were similar between statin-treated children and their unaffected siblings [41]. These findings were then confirmed in other studies, 20 years after the initiation of statin therapy [35]. In view of these observations, the efficacy of statins in pediatrics seems clear; children respond well to HMG-CoA reductase inhibitors and reduce their cumulative estimated atherosclerotic burden: when initiated early (from age 10 years), statins result in delaying the burden from age 35 to 53 years [42,43].

## 4. Safety

Statins are generally very well-tolerated in both adults and children and adverse events are quite uncommon [44]. Although adverse events may be possible, they are often nonserious reversible events, which do not require adjustment in therapy. Moreover, in children receiving statins, the majority of adverse events are as frequent as in those receiving a placebo and they are only drug-related in few cases (e.g., elevated creatine phosphokinase levels after physical activity) [14,34]. Numerous randomized control trials, Cochrane reviews and subsequent meta-analyses consistently established the good short-, medium- and long-term safety of statins in children, especially in those at greater cardiovascular risk (e.g., FH) [28,29,30,41,44,45,46]. Currently, the longest study evaluating the safety profile was performed in children with FH in a 20-year follow-up [35]. Adverse effects primarily include liver toxicity, muscle-related adverse events and abnormalities in laboratory measurements [47]; in addition, statins may have teratogenic effects and may interfere with other drugs.

Liver toxicity: The risk of liver toxicity appears to be particularly low. As with adults, the risk of serious hepatotoxicity is extremely low and no pediatric cases were reported [33,44]. On the other hand, transient elevations in liver enzymes (alanine aminotransferase and aspartate aminotransferase) may occur more frequently (up to 5% of statin-treated children); however, while increased, transaminases did not exceed three times the upper limit of normal or normalized without discontinuation of statin treatment [34]. Interestingly, recent studies did not note any differences in the risk of liver enzymes abnormalities between treatment and placebo groups [41,44].

Muscle-related adverse events: In children receiving statins, the risk of muscle-related adverse events is also particularly low. Myopathy is very uncommon (<0.1%) and no cases of pediatric rhabdomyolysis have been attributed to statins to date [28,44]. Myalgia appears to occur more frequently, although the real extent of the problem remains controversial [14]. In blinded randomized control trials, myalgia was often not associated with elevations in muscular enzymes (creatine phosphokinase); moreover, the frequency of myalgia was lower than that reported in clinical practice, suggesting a “nocebo effect” [48,49]. As with regard to creatine phosphokinase levels, no significant changes (10 times above the upper limit of normal) were found when starting statin therapy and no differences were noted between statin-treated children and a placebo group [28,50]. Although the fear of muscle-related adverse events is a common reason for delaying or not initiating statin treatment, it is important to know that patients often remain asymptomatic and, if muscular symptoms occur, they usually resolve spontaneously without therapy discontinuation [34,50].

Growth, development and cognition: Over time, several concerns have emerged regarding the risk of interfering with the cholesterol synthesis pathway. As mentioned above, in children, cholesterol is essential for cellular functionality, steroid hormones and vitamin synthesis and for adequate growth and development (especially for normal brain development) [51]. Several studies have been published on this issue and most of them agree that statins have no effect on growth and development. When compared to a placebo, there was no convincing evidence that statins may alter growth velocity, cognitive function and educational level, sexual maturation (assessed by Tanner staging and age at menarche), hormone concentrations (e.g., estradiol, testosterone), menstrual cycle length or erectile function [14,28,34,41]. Concerns of statins affecting growth and development were also proven unfounded when statins were used in homozygous FH children as early as 2 years of age [25].

Diabetes mellitus: The risk of developing type 2 diabetes mellitus in children receiving statin treatment is not well clarified, but seems to be lower than in adults (0.2% per year approximately). It is likely that adults have a higher risk of new type 2 diabetes mellitus because of older age, a more aggressive therapy and more frequent comorbidity and concomitant drug use [44]. Interestingly, in a 20-year follow-up study, 1 of the 184 children receiving statins (0.5%) and 2 of the 77 unaffected siblings (2.6%) developed diabetes [35].

Other side-effects: As with adults, mild adverse effects may include hypersensitivity reactions (rash, urticarial), gastrointestinal (dyspepsia, abdominal pain, constipation, diarrhea) and neurological symptoms (headache, dizziness, asthenia). Nevertheless, there is no support of a relationship between statin use and peripheral neuropathy [52]. Almost all of these symptoms resolve with continued use of the statin [14].

Teratogenicity: In accordance with the most recent guidelines for childhood dyslipidemia, statins should be avoided during the preconception period, pregnancy and lactation due to their possible teratogenic effects [53]. However, it is important to note that a recent study systematically reviewed the existing studies on this topic, including almost 1.3 million participants. Very interestingly, the authors concluded that statins were not associated with increased birth defects and that there was no clear evidence that statins were teratogenic [54]. Further studies are therefore needed to investigate this issue.

Drug interactions: As a substrate of cytochrome P450, statins are associated with multiple drug interactions. Competing substrates are able to increase the systemic plasma concentrations of statins and their lipid-lowering effects, as well as the frequency of adverse events [55]. Because of their extensive first-pass hepatic metabolism, drug interaction effects are highest for simvastatin and lovastatin [14]; on the other hand, not being metabolized by a cytochrome P450 isoenzyme, pravastatin is the preferred choice for patients at greater risk of drug interactions [44]. Competing substrates that should be avoided include antifungal azoles, macrolides, cyclosporine, protease inhibitors and grapefruit; instead, dose adjustments are often needed for amiodarone and calcium channel-blockers. Moreover, when receiving statins, patients should also avoid natural remedies containing varying amounts of natural monacolin K (e.g., red yeast rice) [56].

## 5. Indications and Recommendations

Seven commercially available statins are currently available for children and adolescents, being indicated at varying ages and dosages [32]. Table 1 provides a summary of the approved statins in children and adolescents (Table 1).

Rosuvastatin was recently approved for use in children from 7 years old (initially it was authorized for use in children from 10 years old); pitavastatin and pravastatin are indicated in children as young as 8 years old; atorvastatin, fluvastatin, lovastat in and simvastatin may be used in children ≥ 10 years old [31,67]. At present, statins are approved by the Food and Drug Administration (FDA) for children with FH and represent the first-line lipid-lowering drug (statins supplanted bile acid sequestrants) [68]. Patients with FH, due to the underlying genetic disorder, have persistent LDL-C elevations and are more prone to accelerated atherosclerosis and premature ASCVD (high-risk condition) [69]. It is therefore important to early recognize and effectively manage these patients, despite being asymptomatic [6]. Several guidelines and statements provided recommendations for statin treatment in pediatric populations [70,71,72]. According to The National Heart, Lung, and Blood Institute (NHLBI) guidelines and American Heart Association (AHA) Scientific Statement, the average of at least two separate fasting lipid profiles (obtained at more than two weeks, but no more than three months apart) is used to determine which patients are potential candidates for statin treatment [33,69]. Moreover, when evaluating the need and the timing for drug treatment, the presence of secondary causes, familial history of premature ASCVD, additional high-risk factors and risk-modifying conditions should also be considered (Figure 2).

In children with heterozygous FH, guidelines recommend starting statin therapy from the age of 8–10 years: statins should be initiated after secondary causes of dyslipidemia have been excluded and in such cases of persistently elevated LDL-C concentrations despite 3–6 months of lifestyle modifications [63,73,74]. In heterozygous FH children with more severe forms of LDL-C abnormalities, statin treatment may also be started concurrently with therapeutic lifestyle changes [25,31]. On the other hand, in children with homozygous FH, the lipid-lowering therapy should be initiated at diagnosis and often in infancy, although the FDA granted approval for statins from the age of 7 years [63]. Given the most severe lipid abnormalities and the notably increased cardiovascular risk, these patients necessitate an earlier treatment in order to change the natural history of the disease, normalize the atherosclerotic burden in adulthood, and to improve the prognosis [14]. In fact, it is important to remember that the severity of atherosclerosis depends on the lifetime exposure to elevated LDL-C concentrations [1]. In homozygous FH, statins represent the pillar of the treatment despite not being sufficient alone for achieving the LDL-C goals (only 20% of children reach the target) [35]. As with regard to the therapeutic targets, it is noteworthy that there are no evidence-based LDL-C goals for pediatrics: however, when statin therapy is initiated, commonly used targets are:-LDL-C values < 130 mg/dL in moderate-risk patients (heterozygous FH)-LDL-C values < 100 mg/dL or a 50% reduction in LDL-C from baseline values in high-risk patients (homozygous FH) [75,76].

## 6. Prescribing Statin Use in Youth

When statin therapy should be initiated, the decision of a particular statin is mainly a matter of healthcare provider preference, but the choice should also take into account the child’s age, baseline LDL-C values and LDL-C goals [31]. Moreover, the family and patient’s preferences should also be considered, especially if a family member is already taking a particular statin formulation [14]. Statins should be administered once a day, at the lowest available dose: no dose adjustments based on body weight are required [33]. Because most cholesterol synthesis occurs overnight, HMG-CoA reductase inhibitors are typically prescribed with the evening meal or at night [44]. However, this is not always necessary: statins may be taken with or without meals and some of them (e.g., rosuvastatin, atorvastatin), having a long terminal half-life (approximately 19 and 14 h respectively), may be taken at any time [77]. Before initiating statins, it is imperative for clinical practitioners to consult patients and families in order to help to address their hesitancies and concerns regarding the benefits, risks and impacts (e.g., impact on quality of life) of statin therapy. Clinicians should explain the potential need for a lifetime drug therapy and the importance of adopting healthy habits as a synergistic lipid-lowering mechanism [25,78]. Given the possible teratogenicity of statins, all sexually active females must be counselled on having an appropriate contraception [79]; moreover, in view of their pharmacokinetics, a detailed review of potential drug interactions should always be addressed when prescribing HMG-CoA reductase inhibitors [80]. Considering that there are over 30 million patients with FH worldwide (20–25% of whom are younger than 19 years old) and most of them are not diagnosed and/or treated, it is important to promote and increase the knowledge on lipid disorders and their treatment [81,82]. A recent European study, comprising approximately 3.000 heterozygous FH children, showed surprising results: first, there were large between-country differences in the mean age of diagnosis of FH (e.g., 3 years in Greece, 11 years in Belgium). In addition, a significant proportion of the children who were potential candidates for lipid-lowering treatment were not on statin therapy in Europe; in fact, across the European countries the proportion was again very different. This was probably associated with the very different diagnostic strategies and policies used in the different countries [83]. In view of these observations, it is important that clinicians know the current recommendations and be familiar with the characteristics, the dosage and the formulations of at least one of the statins [1].

## 7. Surveillance

As with any new medication therapy, it is important for clinicians to monitor patients taking statins [84]. In particular, the efficacy of treatment, the adherence to therapy and the onset of adverse effects should all be carefully monitored. The most up-to-date guidelines suggest repeating a fasting lipid-profile 4–8 weeks after statin initiation; if adequate LDL-C reduction is observed, a fasting lipid profile should be repeated every 3–6 months (in the first year) and longitudinally every 6–12 months (Figure 3) [33]. If sufficient LDL-C reductions are not achieved, the adherence to therapeutic lifestyle changes and to statin treatment must be assessed [85]. While the adoption of healthy habits may be challenging, the adherence to statin therapy is usually good: according to some studies, 84% of patients took 80% or more of the prescribed medications [35]. When drug compliance is adequate and the tolerance is good, it is recommended to increase the statin dose by one increment (doubling the dose): no dose adjustments based on body weight are needed [14]. However, it is important to known that doubling the statin dose has only modest effects on LDL-C concentrations (resulting in further reduction of approximately 6%) [25]. Moreover, the increased dose may result in more adverse events, although the maximal dose of potent statins for children is between 25% and 50% of the one approved for adults [32]. Only when the combination of maximally dosed statins and lifestyle changes results in an inadequate LDL-C reduction, the addition of a second lipid-lowering therapy (e.g., ezetimibe) should be considered, in accordance with a lipid specialist [69]. It is important to check a fasting lipid profile at each dose and therapy modification or when it is clinically indicated. Current guidelines do recommend assessing transaminases and creatine phosphokinase levels at baseline and to constantly monitor the onset of new symptoms. After initiating statin therapy, it is also suggested to monitor liver enzymes and creatine phosphokinase at 1–2 months and to assess liver enzymes at 3–6 months and periodically thereafter: routine monitoring of creatine phosphokinase is not required because it may result in incidental increases of muscular enzymes that may be related to physical activity [14,33]. Nevertheless, if myopathy symptoms occur, creatine phosphokinase should be measured immediately. When slight increases in liver enzymes or creatine phosphokinase occurs, the discontinuation of statin treatment is not necessary; however, when transaminases and creatine phosphokinase levels increase over 3-fold and over 10-fold, respectively, a prompt discontinuation of statins and the exclusion of other causes of muscular and liver dysfunction is suggested [44]. When children are on statin therapy, it is also advised to monitor growth, development and sexual maturation and it may be reasonable to periodically monitor fasting glucose concentrations (especially in patients at risk of diabetes) [85]. As mentioned above, adolescent females should be counselled on the potential teratogenicity of statins: they should be informed about the importance of contraception and discontinuing statin therapy prior to planned conception (at least 3 months before), pregnancy and lactation [63]. At each visit, it is also recommended to review the potential drug interactions of statins. HMG-CoA reductase inhibitors are typically safe and statin intolerance is uncommon in children; even though side effects occur, it is not contraindicated to reinitiate the previous statin therapy (once laboratory abnormalities and symptoms are resolved). Alternatively, if LDL-C targets were achieved, it is also possible to administer a lower dose of the same statin, the same dose on alternate days or another statin formulation (with different pharmacokinetics) before using a non-statin therapy [14,44].

## 8. Limitations

As mentioned above, due to their possible teratogenic effects, statins should be avoided during the preconception period and pregnancy; it is recommended to cease statin therapy for at least three months prior to becoming pregnant or as soon as pregnancy is recognized. Female patients should also avoid statins during lactation; if they require statin therapy, they should receive direction to immediately stop breastfeeding [14]. Statin contraindications also include children with hypersensitivity to any of its components and patients with active liver disease [44]. Statins should be prescribed with caution in patients with concurrent administration of interfering drugs, predisposing factors for myopathy and patients with chronic kidney disease. In the latter case, dose modification is required and only atorvastatin and simvastatin can be used (being not metabolized in the kidneys) [55].

## 9. Conclusions

Mounting evidence suggests that cardiovascular risk factors, such as dyslipidemia, may be present early on in life. In view of this, and considering that children with lipid abnormalities are more likely to become adults with dyslipidemia, it is extremely important to identify and manage children with lipid abnormalities early (especially those with more severe forms of dyslipidemia). However, in order to achieve these goals, it is important to widely promote the knowledge in this field. The treatment of childhood dyslipidemia has undergone significant changes in recent years: statins are now considered the first-line pharmacologic therapy and the cornerstone of FH treatment during childhood. Statins have increasingly been used in children and adolescents, including those at risk for premature ASCVD (e.g., FH): as in adults, statins effectively improve the lipid profile (mainly by lowering LDL-C), slow the progression of the atherosclerotic process and reduce atherosclerotic burden in adulthood. Current data suggest that the use of statins during childhood may potentially normalize the future cardiovascular risk despite not reaching LDL-C goals. As a consequence, these findings highlight the potential importance of the pleiotropic effects of statins and the utility of precocious treatment when atherosclerosis is reversible. HMG-CoA reductase inhibitors are safe and well-tolerated; as opposed to adults, less aggressive therapy is initiated and less frequent comorbidities are present. Therefore, the risk of adverse events is lower in children than adults. Despite being safe and effective, many children with lipid disorders are not on statin therapy and are not receiving the full potential benefit of adequate lipid-lowering therapies. It is therefore important that clinicians become familiar with statins: this also applies to primary care providers, who are often the first to diagnose childhood dyslipidemia. Lastly, given that most studies were in pediatric FH, more studies are needed to evaluate statin use in children with other risk conditions, such as obesity and diabetes. The increasing prevalence of obesity and diabetes altering the landscape of childhood dyslipidemia combined with dyslipidemia of obesity currently comprise the most frequent phenotype of lipid abnormalities.

## Figures and Tables

**Figure 1 ijms-24-01366-f001:**
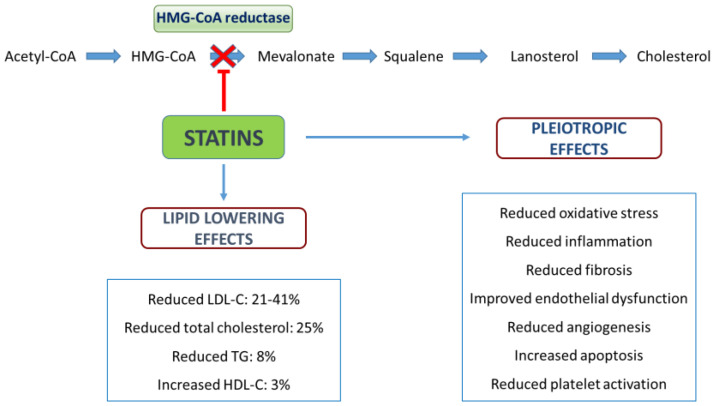
Mechanisms and effects of statins.

**Figure 2 ijms-24-01366-f002:**
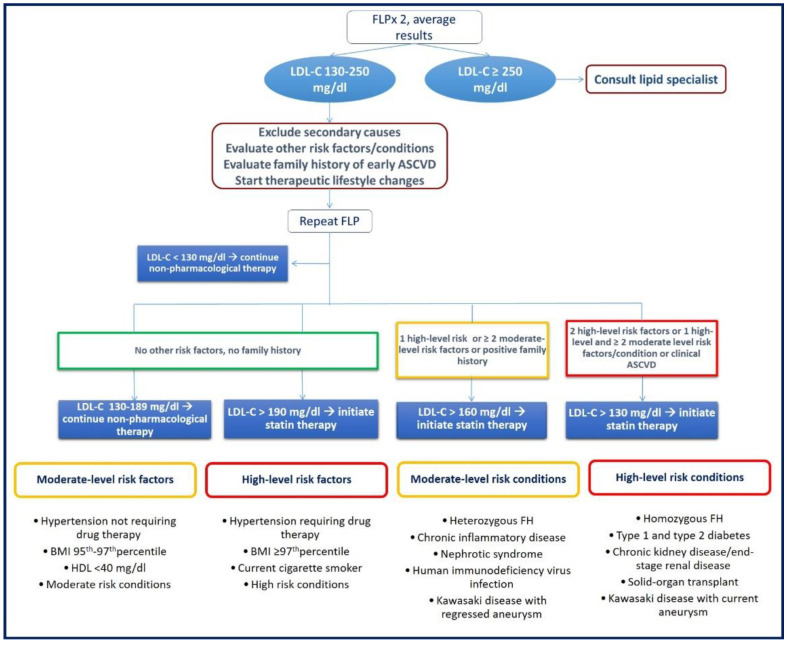
Recommendations for statin treatment in pediatric populations.

**Figure 3 ijms-24-01366-f003:**
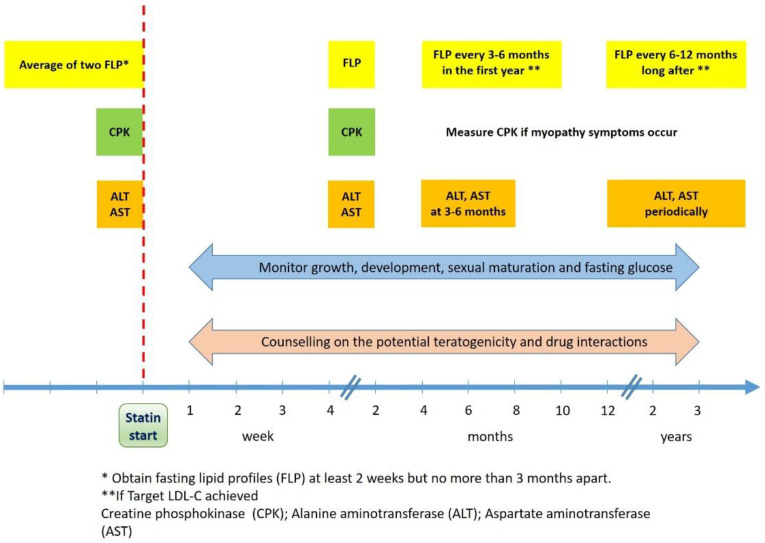
Surveillance scheme in children and adolescents taking statins.

**Table 1 ijms-24-01366-t001:** Commercially available statins for children and adolescents.

Drug	Potency	Pediatric FDA Approvals	Age	Dose	Mean LDL-C % Reduction	Supporting Studies
Atorvastatin	High-potency	Heterozygous FH	10–17 years	10–20 mg/day	40%	McCrindle et al. [57]
Fluvastatin	Low-potency	Heterozygous FH	10–16 years	20–80 mg/day	34%	Van der Graaf et al. [58]
Lovastatin	Low-potency	Heterozygous FH	10–17 years	10–40 mg/day	17–37%	Lambert et al. [59] Stein et al. [60] Clauss et al. [61]
Pitavastatin	High-potency	Heterozygous FH	≥8 years	1–4 mg/day	23–39%	Braamskamp M.J. et al. [62]
Pravastatin	Low-potency	Heterozygous FH	8–18 years	20 mg (8–13 years)	23–33%	Knipscheer et al. [63]
				40 mg (14–18 years)		
Rosuvastatin	High-potency	Heterozygous FH	8–17 years	5–20 mg/day	38–50%	Avis et al. [64]
		Homozygous FH	≥7 years			
Simvastatin	Moderate-potency	Heterozygous FH	10–17 years	10–40 mg/day	31–41%	Couture et al. [65] de Jongh et al. [66]

## Data Availability

Not applicable.

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
