# Peer review of "Statins in Children, an Update"

_ijms, 2023, doi:10.3390/ijms24021366_

Round 1

Reviewer 1 Report

This article is an update and also a reminder of the importance of statins in adults and also in children. It contains information of general interest.
In order to respect the correct structure of a review, I recommend that you also add the methodology by which the article was made.

Author Response

In order to respect the correct structure of a review, I recommend that you also add the methodology by which the article was made. 

  • This article is a narrative review. As such, its purpose is to describe a problem of interest and follow no specified protocol (unlike systematic reviews). This narrative review has no predetermined research question or specified search strategy, only a topic of interest. No standards or protocols guide the review. 

Reviewer 2 Report

This review aims to highlight the efficacy, safety and rationale for the use of statins in pediatrics. The commenting of the review topic is complete, the content of the review is significant, the presentation is pretty good and relevant to the field.

HMG-CoA is a very important intermediate of cholesterol synthesis and cholesterol is relevant to hormone biosynthesis. Statins, as HMG-CoA) reductase inhibitors, may interfere with many areas of biosynthesis. Side effects should be taken into consideration.

As a review, complete of topic is important. Can you add some side effects on neurological, digestive system, sexual problems, etc.?

There are minor revisions may need:

1.    Line 19 and line 305: Is the consul typo error of counsel?

Author Response

As a review, complete of topic is important. Can you add some side effects on neurological, digestive system, sexual problems, etc.?

--> Neurological and digestive side effects were added. Moreover, it was clarified that menstrual cycle length and erectile function are not affected by statins treatment.

 There are minor revisions may need:

  1. Line 19 and line 305: Is the consul typo error of counsel?

--> Spelling mistakes have been corrected.

Reviewer 3 Report

The authors reviewed the state-of-the-art lipid-lowering therapy in infant dyslipidemia with a focus on statins. After explaining the cholesterol metabolism and the treatment mechanism of statins, the authors depict the current knowledge on statin therapy in children and adolescents. The review shows that statins in infancy are safe as well as efficient, which should encourage clinicians to a prescription if necessary. Furthermore, the authors reviewed indications for statins as well as the surveillance of the prescribed therapy, giving precise recommendations on when to follow-up for example LDL-C values and, equally important, which values do not need a regular follow-up. In conclusion, this review is an excellent summary of the current knowledge on statin therapy in infancy. Therefore, the reviewer has only several minor concerns regarding orthography listed in the following on a point-to-point basis:

1.    Page 1, Line 19: … to consult patients … (?)

2.    Page 2, Line 84f.: very-low-density lipoproteins (VLDL-C), intermediate-density lipoproteins (IDL-C), low-density lipoprotein (LDL-C), high-density lipoproteins (HDL-C)

3.    Page 3, Line 119: … in adults: although their … (maybe “besides” instead of “although”)

4.    Page 3, Line 123: … studies and meta-analyses …

5.    Page 3, Line 124: … most studies were in the pediatric familial hypercholesterolemia (FH), … ïƒ  maybe: … most studies were conducted in patients with pediatric familial hypercholesterolemia (FH), …

6.    Page 3, Line 131: … it is important to remember that …

7.    Page 4, Line 142: … the use of statins in adults significantly reduces cardiovascular morbidity …

8.    Page 4, Line 142: … and this also applies to those patients with FH. In adults with FH…

9.    Page 4, Line 167: … when were compared with non-FH …

10. Page 4, Line 186: … follow-up of 

11. Page 7, Figure 2: Concerning the two blue rectangles on the right: Would “LDL-C > 160 mg/dl ïƒ  initiate statin therapy” (left) and “LDL-C > 130 mg/dl ïƒ  initiate statin therapy” (right) be correct and easier to understand?

12. Page 7, Figure 2: Do the moderate- and high-level conditions really only apply to the right, red-framed rectangle, as they are only mentioned there?

13. Page 7, Line 276: … despite a 3-6 months …

14. Page 8, Line 305: … to consult patients … (?)

15. Page 8, Line 314: … treated, it is …

16. Page 9, Line 345: … at each dose and therapy modifications or when it is clinically indicated.

17. Page 9, Line 348: … monitor liver enzymes …

18. Page 9, Line 354: … slight increases … occurs …

19. Page 9, Line 356: … discontinuation of statins is suggested and … dysfunction is suggested.

20. Page 9, Line 358: … it may be reasonable …

21. Page 9, Figure 3: Alanine aminotransferase (ALT)

22. Page 9, Line 372: 8. Conclusions

Author Response

--> Spelling mistakes have been corrected and figures have been modified.

Reviewer 4 Report

The authors have briefly covered the role of statin and its application on children in the manuscript.

However, I would suggest including a table with detailed information on different studies on statin use in children, age, type of statin, and effect. 

In addition, I would recommend including a limitation section on the use of statins in children. Some of the points are mentioned in surveillance. 

Author Response

However, I would suggest including a table with detailed information on different studies on statin use in children, age, type of statin, and effect.

--> Table 1 was modified by adding the mean effect of statins on LDL and reference to the supporting studies.

In addition, I would recommend including a limitation section on the use of statins in children. Some of the points are mentioned in surveillance.

-->  Table 1 Limitation section has been added